# Horizontal face information is the main gateway to the shape and surface cues to familiar face identity

Helene Dumont[1]*, Alexia Roux-Sibilon[1,2], Valérie Goffaux[1,3]

**1** Psychological Sciences Research Institute (IPSY), UC Louvain, Louvain-la-Neuve, Belgium, **2** Université Clermont Auvergne, CNRS, LAPSCO, Clermont-Ferrand, France, **3** Institute of Neuroscience (IONS), UC Louvain, Louvain-la-Neuve, Belgium

* helene.dumont@uclouvain.be

**Data Availability Statement:** All data and analysis codes are help by openicpsr, in the project named Inversion_Negation_HorizontalTuning. They can be found at https://www.openicpsr.org/openicpsr/project/208541/version/V1/view.

## Abstract

Humans preferentially rely on horizontal cues when recognizing face identity. The reasons for this preference are largely elusive. Past research has proposed the existence of two main sources of face identity information: shape and surface reflectance. The access to surface and shape is disrupted by picture-plane inversion while contrast negation selectively impedes access to surface cues. Our objective was to characterize the shape versus surface nature of the face information conveyed by the horizontal range. To do this, we tracked the effects of inversion and negation in the orientation domain. Participants performed an identity recognition task using orientation-filtered (0˚ to 150˚, 30˚ steps) pictures of familiar male actors presented either in a natural upright position and contrast polarity, inverted, or negated. We modelled the inversion and negation effects across orientations with a Gaussian function using a Bayesian nonlinear mixed-effects modelling approach. The effects of inversion and negation showed strikingly similar orientation tuning profiles, both peaking in the horizontal range, with a comparable tuning strength. These results suggest that the horizontal preference of human face recognition is due to this range yielding a privileged access to shape and surface cues, i.e. the two main sources of face identity information.

## Introduction

Among the myriad of essential social cues conveyed by the human face (e.g., age, emotion, familiarity...; [1], identity plays a crucial role in human social organization (e.g., cooperation, kin recognition). The ability to recognize identity in a species is supported by the presence of distinctive identity signals [2]. The study of Sheehan & Nachman (2014) [3] suggests that human social evolution maximized the human face phenotypic and genetic diversity. For instance, this is shown by a larger variation of length and distance among facial features than among non-facial features (i.e. someone with big hands is more likely to have big fingers, but someone with a big nose in not more likely to have big eyes or mouth).

The important function of human face identification relies on specialized visual mechanisms and brain regions (e.g., [4–7]). Humans reach very high identification performance

**Funding:** This work was supported by the FNRS (Fond National de la Recherche Scientifique https://www.frs-fnrs.be/en/) ASP grants n°: 1.A.681.22F granted to HD and the Excellence of Science grant HUMVISCAT- 30991544. VG is an F.R.S.-F.N.R.S. Research Associate. ARS was supported by a FSR Postdoc Fellowship grant of UCLouvain (Université Catholique de Louvain https://uclouvain.be/en/index.html). The funders did not play any role in the study design, data collection and analysis, decision to publish, or preparation of the manuscript.

**Competing interests:** The authors have declared that no competing interests exist.

especially in the case of naturally familiar faces, which can be recognized even with bad quality images or videos and drastic changes in appearance [8, 9]. Such tolerance of familiar face recognition is impressive as variability in the appearance of the same face across pictures taken under different conditions (illumination, viewpoint, expression, camera) often exceeds variability between pictures of different identities taken under the same conditions [8, 10, 11]. The visual mechanisms and information allowing the tolerance of familiar face identification are not yet well understood and are extensively researched in vision sciences [12–15].

Recent evidence showed that the information that drives human face identity recognition is optimally conveyed by an orientation range centred around the horizontal axis. Dakin & Watt (2009) [16] filtered familiar face images in the Fourier domain to only keep the information contained in a narrow orientation band. They found that identity recognition performance describes a bell-shaped curve peaking in the horizontal range and reaching its minimum in the vertical range. Such horizontal tuning of face identity recognition has been documented in several studies, and replicates across various methodologies, e.g. orientation filtering [16–20], oriented noise masking [19, 21] and oriented phase randomization [22]. It is present from three months of age on [23] and develops over lifespan [17, 24, 25]. The perception of face emotional expressions has also been found to be horizontally-tuned [26, 27]. This horizontal tuning is correlated to face recognition performance in participants exhibiting normal recognition skills [21, 28, 29] and is stronger for the recognition of familiar faces [30], suggesting a link between horizontal tuning and genuine and specialized face identity recognition.

This link between horizontal tuning and face-specialized identity processing is further supported by evidence that the former is disrupted by picture-plane inversion [18–22]. Inversion, while largely preserving image content, greatly disrupts face recognition. Faces shown upside-down are harder to recognize than faces shown in a canonical upright position, and this decrease in recognition performance affects faces disproportionately more than other mono-oriented objects or scenes [31–36]. Similarly to inversion, another image manipulation known to affect the recognition of faces more than other objects is contrast-polarity reversal (referred to as negation in this paper; [37, 38]).

The disproportionate effects of inversion and negation indicate the existence of visual mechanisms tuned to the natural statistics of the human face. Indeed, over evolution and lifespan, the human visual system presumably builds an internal representation of how faces appear in everyday life, i.e., usually in upright position, lit from above [39] and depicting characteristic contrast relationships (i.e., an alternation of light and dark, with facial features relatively darker than surrounding skin; [40–42]). The typical vertical contrast alternation of the human face is thought to be conveyed by the horizontal range of the face image [16, 43]. Such natural statistics are reversed in space and polarity in inverted and negated faces, respectively, which likely explains the drastic influence of these manipulations on face recognition [16]. Past evidence showed that inversion strongly weakens the horizontal tuning of human face identity recognition, with a reduction in the amplitude of the horizontal peak and a doubling in bandwidth [20]. However, whether and how the the polarity reversal of the face contrast alternance caused by negation influences the orientation tuning profile of human face identification remains elusive.

While inversion and negation both alter horizontal face structure, they do so in very distinct ways. Indeed, while inversion flips the edge and shape properties of the face upside-down, negation reverses luminance gradient polarity but preserves edge and shape information. Past behavioural [44–46] and neural [47, 48] evidence further indicates that they disrupt distinct aspects of face processing. Inversion is commonly thought to disrupt the so-called holistic processing (i.e. perceiving the face as a whole; [5, 34, 49]) of face shape properties [49–51]. On the other hand, negation has been proposed to disrupt the access to face surface cues such as

pigmentation [44, 52, 53], and shape-from-shading cues [44, 54, 55]. Although some recent studies found little to no effect of negation on holistic processing [45, 56], others reported that disruption of surface cues with negation could also result in a disruption of holistic processing [57]. Overall, while some studies indicate some functional overlap between these effects, inversion and negation are generally thought to disrupt relatively distinct aspects of face information: shape and surface, respectively.

Our study characterizes the orientation profile of inversion and negation effects on human face recognition in order to infer the shape versus surface nature of the information contained in the horizontal range of the face stimulus. We used familiar face stimuli to tap into genuine and specific face recognition mechanisms [8, 30]. We measured the difference between the orientation tuning profile for natural faces (faces in natural upright orientation and contrast polarity) and that for (i) inverted faces and (ii) negated faces in the same participants. A difference in the horizontal tuning profile of the inversion and negation effects would indirectly point to a differential contribution of this orientation range to shape and surface processing. Alternatively, finding that inversion and negation similarly modulate recognition orientation profile would indicate that the horizontal range conveys either the information driving shape and surface cue processing of the face image, or a more fundamental aspect of the natural statistics of the upright face coming in play prior to the presumed functional dissociation of shape and surface cue processing.

## Material and methods

### Subjects

40 healthy young adults were recruited (from 15/06/22 to 12/12/22) for the experiment. Only 21 of them, eight males, aged 23.6 (± 4.5) years old, recognized the requested number of celebrities and completed the experiment in exchange for (i) course credits (1 credit per hour of testing) for participants recruited via a psychology course, or (ii) monetary compensation (10 euros per hour of testing) for participants recruited through a Facebook ad. All participants gave their written, informed consent, based on a short explanation of the protocol before starting the experiment. At the end of the experiment (usually after three sessions) participants received detailed information regarding the experiment and had the opportunity to ask any question they might have had during a debriefing. Visual acuity was tested with the Landolt C test of the $FrACT_{10}$ v1.0 [58] and all logMAR scores indicated normal or corrected to normal vision (logMAR<0). Face perception abilities were measured with a computerized Benton Facial Recognition Task [59] and all scores indicated normal face perception (score = 46.6 ± 3.6). The experimental protocol was approved by local ethical committee (Psychological Sciences Research Institute, UCLouvain).

### Stimuli

In order to compare the influence of inversion and negation on the horizontal tuning of face identity recognition, we measured participants' identification performance with upright-positive (i.e., 'natural'), inverted, and contrast-negated stimuli that were either full-spectrum, or orientation-filtered in Fourier domain.

Stimuli consisted of greyscale, 302 x 309 pixels, full front or close to full front, faces images of contemporary white American or British famous male actors (18 actors, four pictures each) selected from the Facescrub dataset [60] and a Google search for free to use images. Images were carefully selected to exclude any distinctive accessory (glasses, hat, . . .) and to have a neutral or close-to-neutral facial expression. Eighteen celebrities were selected from a larger set of actor images with an online pilot running on Pavlovia (Open Science Tools, Nottingham, UK)

**Orientation Filter:**

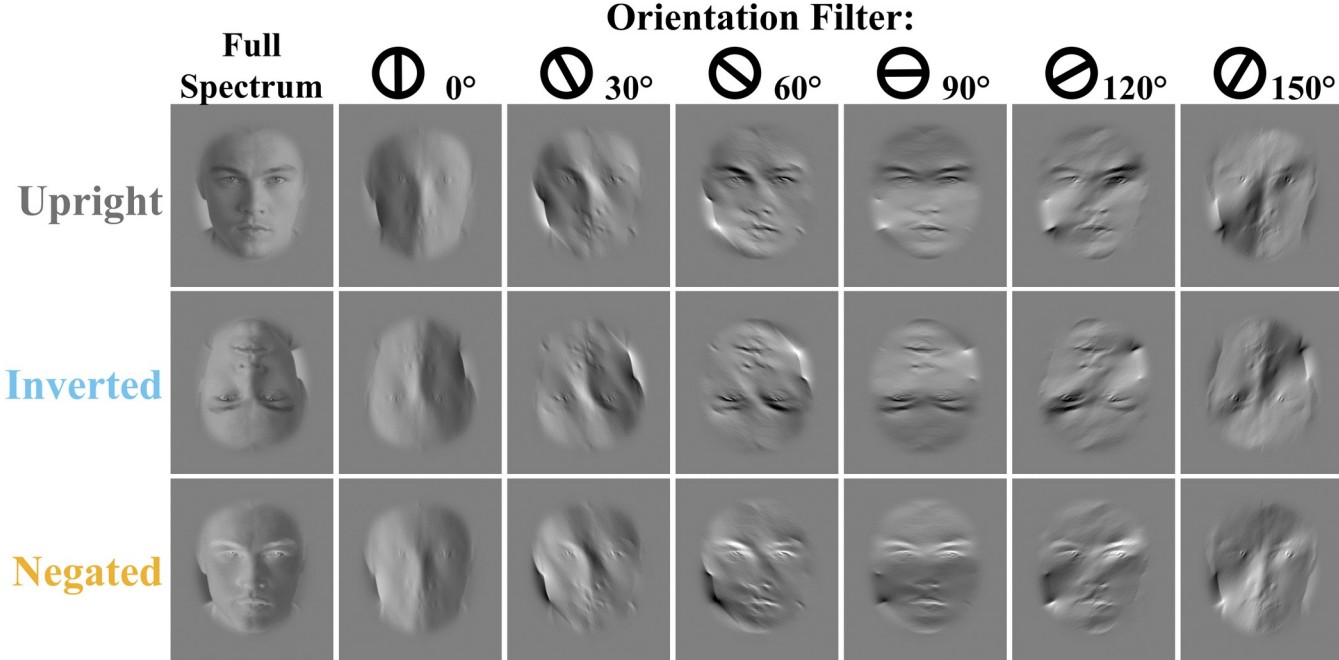

**Fig 1. Example of the stimulus conditions.** Each column represents the orientation-filter condition starting by non-filtered images and going in steps of 30°
from 0° to 150° from left to right. Each row represents the stimulus type (natural, inverted, negated).

to ensure that they were recognized by most participants. We only kept the four best recognized images for each recognized actor (see S1 in S1 File), resulting in a total of 72 images.

Images were converted to greyscale and rotated to align the eyes with the horizontal meridian. We then cropped and resized each one using forehead and chin as points of reference to match face size across all images. The resulting images were padded with 100 pixels that were mirroring the face image, to prevent border artefacts during filtering. Next, images were normalized to obtain a mean luminance and root-mean square (RMS) contrast of 0 and 1 respectively and fast Fourier transformed, with the amplitude spectra multiplied with six wrapped Gaussian filters centered on 0° to 150° in steps of 30°, where 0° is vertical and 90° horizontal [16, 19, 20]. The Gaussian filters' standard deviation was set at 25° to approximately match the standard deviation of the horizontal tuning of the identity recognition performance in upright unfamiliar faces previously documented by Goffaux & Greenwood (2016) [20] and were close to those used by Dakin and Watt (2009; 23°). This broad bandwidth resulted in some overlap between adjacent orientation ranges but was chosen to ensure that recognition was possible even in the most difficult condition (i.e. inverted and 0° filter). The proportion of clipped pixels in an image (i.e., pixels with an intensity below 0 or above 1, on a [0,1] scale) was below 3%. Mean luminance and RMS contrast of all images were set to the mean luminance and RMS contrast of the image set (0.5 and 0.12 respectively). A grey elliptical aperture was created to match the shape of an average image of all the faces in the set, and blend with the background of the experiment. It was blended with each image (filtered and non-filtered), after RMS equalization, to cover most of the background, hair and the neck (Fig 1). Gaussian white noise masks were also designed with 1/f noise, and global luminance and RMS contrast set to match those of the face images. We used Matlab R2014a for these image processing steps.

In the experiment, images were displayed naturally, inverted or negated. Inverted images were presented using a 180° clockwise rotation on Psychopy. Negated images were made prior

to the experiment, using Matlab by reversing contrast values, after filtering, luminance and contrast equalization but before the elliptical aperture was blended with the picture.

## Procedure

Participants did the experiment on a VIEWpixx monitor (VPixx Technologies Inc., Saint-Bruno, Canada) with a 1920 × 1080 pixels resolution and a 70Hz refresh rate. They used a head-and-chin-rest to support their head and maintain a viewing distance of 60 cm. Face images presented at this distance subtended a visual angle of 9˚ by 10.4˚, which simulates an observer distance ideal for holistic processing [61]; masks had a diameter of 11˚. The experiment lasted for three one-hour sessions. We used Psychopy2021.2.3 to code and run the experiment.

During a prescreening phase, each participant selected 10 actors that they were familiar with (see S1 in S1 File for details). To do so, participants performed a 5-alternative forced choice identity recognition task with upright full spectrum images (four images for each of the 18 actor identities as selected based on the online pilot ran with a different participant sample; see Stimuli). Each image appeared on the center of the screen for 500ms followed by the names of five actors: the name of the actor corresponding to the image and four other names, randomly taken out of the 17 remaining names. Participants were instructed to select the name corresponding to face identity. An actor's identity was selected only if at least three out of the four images were recognized. Once 10 familiar identities were selected, the four images of each identity were displayed again on the monitor for five seconds. With this procedure, the selection of actors was different for each participant and adapted to their knowledge of the identities, thus ensuring a recognition performance rooted into natural familiarity during the main experiment [8]. In the next training, and main experimental trials, three out of the four images per actor were used as stimuli. The remaining images were presented in-between blocks, to remind participants of unmodified selected identities without exposing them with actual stimulus.

Participants performed an identity face-matching task (Fig 2). A trial started with a white fixation dot on a grey background. The first face image (probe) was presented during 500ms followed by a mask presented for 200ms and a second face image (target) for 400ms. Participants had to decide whether faces in the just-seen-pair depicted the same identity or different identities with a keypress ('S' for same identity or 'L' for different identity). Participants had up to 4400 ms to respond from target onset. After target offset, the fixation dot turned black. When an answer was recorded, the fixation dot briefly turned green for a correct answer and red for an incorrect one. The following trial started after a random duration interval (ranging from 750 to 1350ms). Identity was matching in 50% of the trials. Pairs of each trial were composed after the prescreening phase for each participant (30 pairs for *same* and 30 pairs for *different* trials), and replicated across conditions. Importantly, in the case of *same* trials, both images were always two different images of the same identity, ensuring participants were recognizing identity and not merely matching images. Both images of a trial were presented under same stimulus type and orientation-filter condition. During the piloting phase of this study, we noticed difficulties for participants to reach above chance accuracy when conditions were fully randomized. We thus decided to block stimulus type to facilitate task performance.

As our task involved recognition of face images with orientation filtering, coupled with either inversion or negation, participants first started with a training phase. They were familiarized with the task with increasing levels of difficulty, to ensure they would reach stable and sufficient performance before gathering data. The training phase was divided in three steps of increasing difficulty. In the first training step, participants did the face identity matching task

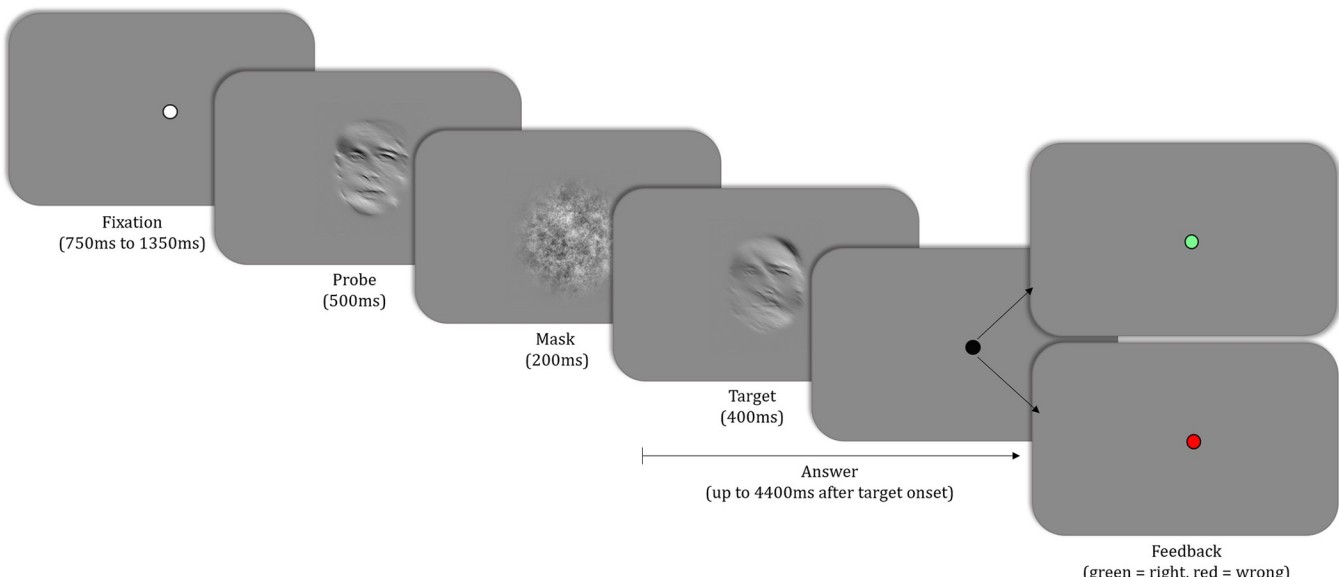

**Fig 2. Example *same* trial sequence of the main experiment.** Each trial starts with a fixation of random duration (between 750ms to 1350ms). A probe face is then presented for 500ms, followed by a 200ms mask and a target face for 400ms. Participants can answer from target onset and up to 4000ms after target offset by pressing either 'S' for same identity or 'L' for different identity. The fixation dot briefly turns green after a correct answer and red after an error.

with natural, full spectrum face images. Once participants reached an accuracy > 80%, training went on to the second step. In the second step, face images were either natural, inverted or (contrast-)negated and full spectrum, and presented per Stimulus type in 14-trial blocks. Blocks were grouped by three, one for each Stimulus type, and scores were calculated over these groups of three blocks. Inside each group of blocks, the order of blocks (i.e., of Stimulus type) was set randomly. Once participants achieved a mean score > 75%, training went on to the third training step which was exactly the same as the second, except that within each block (natural, inverted or negated), stimuli were presented both filtered in orientations and full spectrum (i.e., two trials of each of the six orientations, from 0˚ to 150˚, and two full spectrum trials). To pass this last part of training participants needed to reach a mean score of at least 75% (calculated over three-blocks groups with the three Stimulus type conditions), twice in a row. This ensured participants had good comprehension of the task and were able to score well above chance level (50%) before starting the testing phase.

The main experiment consisted of 90 blocks (30 blocks per Stimulus type). Each block was composed of 14 trials: two trials of each orientation filter plus two full spectrum trials. This resulted in a total of 60 trials per condition (i.e., combination of each Stimulus type and Orientation filter), 420 trials per Stimulus type, and 1260 trials overall. With 30 images (three per preselected identities), three Stimulus type (i.e., natural, inverted, negated) and seven Orientation Filter (i.e., 0˚ to 150˚ in steps of 30˚ and full spectrum) conditions, there was a total of 630 images, each repeated four times (twice in *same* trials twice in *different* trials) throughout the experiment.

Stimulus type was blocked while Orientation filter varied randomly within blocks. Blocks were presented three by three, one per Stimulus type, in a randomized order, with a short break in-between (of a minimum duration of five seconds) and a light tone indicating the beginning of a new block. Longer breaks (of at least 30 seconds) happened every three blocks. Longer breaks started with a score informing participants on their performance over the last three blocks and the number of remaining blocks; and ended with the (non-filtered) picture of each selected actor that was not used during the experimental trials, along with their name.

## Data analysis

Through the comparison of inversion and negation effects on the Orientation filter dependence of identity recognition performance, the purpose of this study was to understand the extent to which the horizontal tuning of face identification reflects its contribution to the processing of shape versus surface cues.

During the experiment accuracy was recorded for each participant. All participants scored above chance level. We calculated the mean response time and standard deviation of each participant and considered outlier every answer with a response time faster than *mean – 2.5\*sd* or slower than *mean + 2.5\*sd*, which resulted in a loss of 3.3% of all answers.

From accuracy data, we computed a sensitivity index (*d'*) to assess participants' performance in bias-free manner (accuracy was also analyzed and produced very similar results–this analysis can be found in S6 in S1 File). We computed a *d'* for each combination of Stimulus type and Orientation filter (with a *same* answer in the presence of same identity considered as a *hit*, a *different* answer in the presence of different identities considered as a *correct rejection*, a *same* answer in the presence of different identities considered as a *false alarm*, and a *different answer* in the presence of the same identity considered as an *omission*). Following the log-linear correction [62], we added 0.5 to the hits and false-alarm counts, and calculated the *d'* value for each subject in each condition and filter. The normalization of dependent variables is common practice when using complex hierarchical models, as it facilitates model convergence (Stan Development Team, 2023). Sensitivity was thus normalized with a Z transform using this formula:

$$normd(x) = \frac{d\prime(x) - \mathrm{mean}(d'(x))}{\mathrm{sd}(d'(x))}$$

where, for a given combination of Orientation filter and Stimulus type *x*, *normd'* is the normalized *d'* output, *d'* is the non-normalized sensitivity value, mean(*d'*) is the group-level mean *d'*, and sd(*d'*) is the group-level standard deviation of *d'*.

Performance at the 0˚ filter was duplicated to have circular filter values from 0˚ to 180˚.

First, to ensure participants scored higher for natural stimulus type, which we use as baseline to calculate inversion and negation effects, average performance (across Orientation filters, excluding the full spectrum trials) in each stimulus type condition was compared to the others using a Bayesian general linear mixed model (GLMM), using participants as a random effect, to consider inter-subject variability in the performance estimations for each condition, with this equation:

$$normd \sim \text{Stimulus type} + (\text{Stimulus type}|\text{Subject})$$

We ran the model with default *brms* priors with four Markov chain Monte Carlo (MCMC) with 6,000 iterations each (2,000 warm up, resulting in 16,000 iterations after warmup), and evaluated the convergence of the chains with diagnostic values such as number of divergent chains post warmup, Rhat statistics and effective sample size.

As expected, the GLMM indicated that participants performed better in natural condition than in inverted and negated conditions (Fig 3).

Recognition performance for natural face images described a broad Gaussian centered around 90˚ and was modelled using a Bayesian Gaussian mixed model (Fig 3). The model is similar as the one used by Goffaux & Greenwood (2016) [20] and described in the next section. Results of this analysis are reported and interpreted in S4 in S1 File.

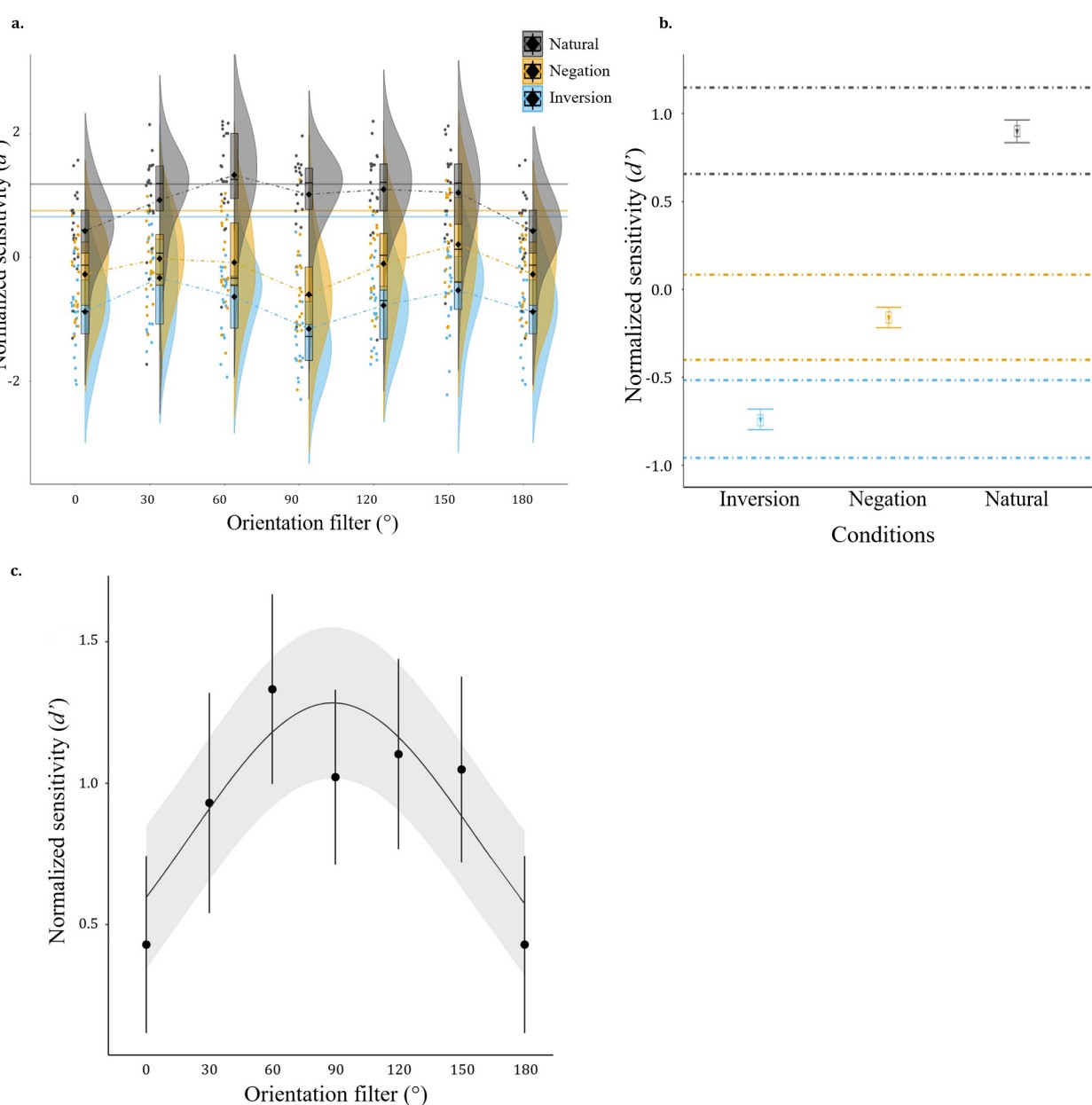

**Fig 3. a. Group-level normalized sensitivity (*normd'*) for identity recognition as a function of Orientation filters.** The black diamonds represent the mean performance for each condition and the boxplots represent the 25%, 50% (median) and 75% quantiles. Each point is one participant's recognition performance for an Orientation filter and Stimulus type. The half violins represent the distribution of the data for each condition. The full lines represent the full spectrum performance for each condition. **b. Predictions from the Bayesian general linear mixed model comparison of Stimulus type.** The non-overlapping 89% credible intervals (CrI) of model predictions indicate a substantive performance difference between each stimulus type. The triangles and error bars represent the group mean and standard error of the data in each condition. The square is the mean in each condition given by the model prediction, and the doted lines are the 89% CrI. **c.** Population-level averaged recognition performance for natural images and Bayesian Gaussian mixed model predictions. The dots are the population-level averaged normalized sensitivity with the 95% CrI in error bars. The model mean posterior prediction is drawn in black (line) and the 89% CrI of the predictions are represented as the faded ribbons.

Next, the magnitude of the inversion and negation effects was calculated in each subject and for each Orientation filter by a subtraction method:

$$Inversion\ effect = \text{Natural } normd' - \text{Inversion } normd'$$

$$Negation\ effect = \text{Natural } normd' - \text{Negated } normd'$$

As can be seen on Fig 4, the magnitude of inversion and negation effects as a function of the Orientation filter described a bell-shaped curve peaking around 90˚ (horizontal range) and decreasing towards 0˚ and 180˚ (vertical range). We used a Bayesian non-linear mixed model to fit a Gaussian function to our data. This enabled the comparison of the effects of inversion and negation along the four parameters of the Gaussian tuning function: (i) Peak Location, in degrees: orientation at which the estimated effect is highest; (ii) Standard Deviation: sharpness of the effect's tuning around the peak orientation; (iii) Peak Amplitude: difference between the effect at its highest (i.e., at the estimated peak orientation) and at its lowest; and (iv) Base Amplitude: minimum size of the effect (i.e., at the orientation where effect estimate is the lowest). The mixed dimension of the model allowed to consider the inter-subject variability during the parameters estimation by setting it as a random effect. The inversion and negation effects were fitted with the following Gaussian equation [20]:

$$deffect \sim Base\ Amplitude + Peak\ Amplitude * e^{\frac{(Orientation\_filter - Peak\_Location)^2}{2Standard\_Deviation^2}}$$

We estimated the four Gaussian parameters (see Fig 4B) with the following linear regression models:

Peak Location ~ 1 + Stimulus type + (1 + Stimulus type | Subject)
Standard Deviation ~ 1 + Stimulus type + (1 + Stimulus type | Subject)
Base Amplitude ~ 1 + Stimulus type + (1 + Stimulus type | Subject)
Peak Amplitude ~ 1 + Stimulus type + (1 + Stimulus type | Subject)

For each of these regression models, prior distributions were defined for intercept (i.e. 1), slope (i.e. the effect of Stimulus type), and random effect of Subjects. Priors for the intercepts of the linear regression models were normal distributions, whose means were chosen based on theoretical expectations. For example, the prior for Peak Location intercept was a normal distribution centered on 90˚, as we expected the effects of inversion and negation to peak for horizontally oriented filters. Priors for slopes of the linear regression models were also normal distributions, whose means were 0 (i.e., the most probable *a priori* was that there would be no difference between inversion and negation effects). Precise values of the standard deviations of prior normal distributions were chosen in a compromise between allowing MCMC chains to converge and keeping priors as broad, thus uninformative, as possible, to not influence the model toward a specific estimation. Finally, priors for random effects were exponential distributions (i.e. positive values only, to specify that there cannot be less than zero variance between subjects; see Table 1 for details about the prior distributions used in the model).

The model was fitted with four MCMC with 6,000 iterations each (2,000 warm up, resulting in 16,000 iterations after warmup). The fit of the model was evaluated with convergence of MCMC chains thanks to diagnostic values such as the number of divergent chains post warmup, Rhat statistics, effective sample size and number of divergent chains. The estimated parameters were compared between inversion and negation effects using their 89% credible intervals (CrI) to test for a "significant" difference.

Since the parameter estimates of the effects of inversion and negation were strikingly similar, we explored whether the two effects were functionally linked at the individual level. We

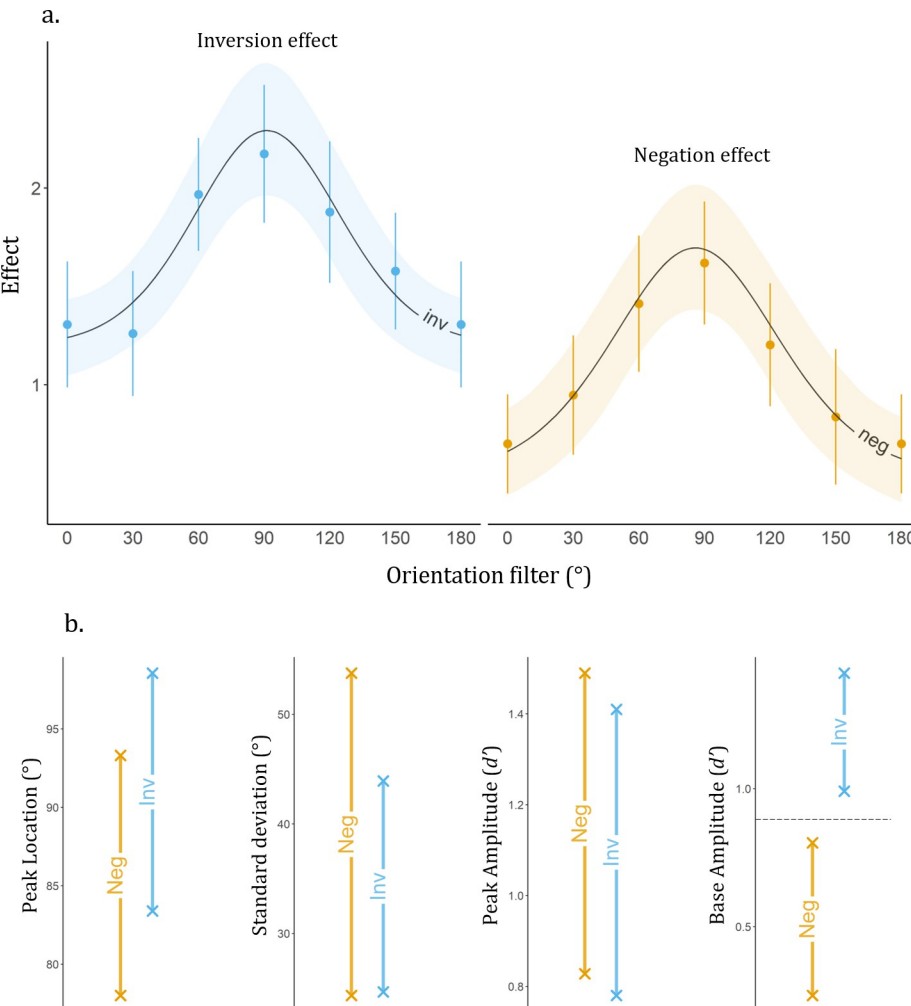

**Fig 4. a. Population-level averaged inversion and negation effects and Bayesian non-linear mixed model predictions as a function of Orientation filter**. On the y axis, inversion and negation effects are calculated at the individual level by subtracting performance (*normd'*) in the Inverted and Negated condition from performance in the Natural stimulus type condition. The dots are population-level mean effects of inversion (blue) and negation (yellow), with the 95% confidence interval in the error bars. The model mean posterior predictions for each Stimulus type are drawn in black (line) and the 89% credible intervals (CrI) of predictions are represented as the faded ribbons. Both Gaussians peaked in the horizontal range and decreased towards the vertical orientation. This shows that inversion and negation effects on identity recognition both culminate in the horizontal range. **b. 89% CrI of the four Gaussian parameters posterior predictions distribution.** A difference is significant when the CrI of the parameters of the inversion effect (blue) and negation effect (yellow) do not overlap. The CrI of Peak Location, Standard Deviation and Peak Amplitude of inversion and negation effect overlapped, indicating highly similar model estimates for both effects. However, there was a significant difference in the estimates of the base amplitude for inversion and negation effects, with a higher CrI for the inversion effect, indicating that the effect of inversion was overall larger that the effect of negation.

ran Pearson correlations between the individual parameters estimates of the inversion and the negation effects for each of the four Gaussian parameters. These results would give us insight on whether inversion and negation disrupt face identification in a similar way.

In Bayesian mixed-effects analysis, "population-level" includes all participants (fixed effects), and "group-level" is used to describe the sub-categories of the multilevel analysis (random effects), here the group-level represents each participant.

**Table 1. Priors of Bayesian non-linear Gaussian models.**

| Parameter of non-linear Gaussian model | Peak Location | Standard Deviation | Peak Amplitude | Base Amplitude |
|---|---|---|---|---|
| Prior for Intercept | normal(90, 20) | normal(35, 20) | normal(1.5, 1) | normal(1.5, 1) |
| Prior for slope (effect of Stimulus Type) | normal(0, 20) | normal(0, 30) | normal(0, 1.5) | normal(0, 1.5) |
| Prior for random effects | exponential(0.1) | exponential(0.1) | exponential(0.1) | exponential(0.05) |

Normal prior distributions were set on parameters of linear regressions (intercept and slope) and exponential distributions were set on parameters estimating random effects.

The analysis was done using R 4.2.2. We used the *brms* package [63, 64] to fit all Bayesian models. All data and analysis codes can be found at https://www.openicpsr.org/openicpsr/project/208541/version/V1/view.

## Results

At a descriptive level, normalized *d'* was higher across all orientations for natural than negated faces, and higher for negated than inverted faces (see Fig 3A and S1 Fig in S1 File).

Bayesian GLMM comparison of Stimulus type showed significant differences in recognition performance between Stimulus type conditions: the 89% CrI of GLMM predictions of the mean performance of the three conditions do not overlap (Inverted CrI = [-.958; -.516], Negated CrI = [-.4; .084], Natural CrI = [.657; 1.148]; see Fig 3B).

Population-level inversion and negation effects as a function of Orientation filter and their Gaussian model are shown in Fig 4A. Inversion and negation effects both showed a Gaussian tuning profile centered on the horizontal range (see Table 2 and Fig 4). Indeed, while inversion and negation disrupted identity recognition across all orientations, both effects peaked in the horizontal range (mean and CrI Peak Location in degrees; inversion: m = 91.52, CrI = [83.40; 98.56]; negation: m = 85.99, CrI = [78.01; 93.31]). The amplitude of the horizontal peak was comparable across inversion and negation effects (mean and CrI Peak Amplitude in *d'effect*; inversion: m = 1.11, CrI = [.78; 1.41]; negation: m = 1.18, CrI = [.83; 1.49]). The Standard Deviation of the Gaussian tuning functions was also similar across effects (mean and CrI in degrees; inversion: m = 34.06, CrI = [24.66; 43.92]; negation: m = 39.39, CrI = [24.34; 53.77]). The only difference was found at the level of the Base Amplitude of the Gaussian functions relating inversion and negation effects to Orientation filter (mean and CrI in *d'effect*; inversion: m = 1.20 CrI = [.99; 1.42], negation m = .53, CrI = [.25; .80]). The fact that the base amplitude was larger for inversion than negation (and that this was the only difference between the two effects) shows that inversion impaired recognition performance overall more than negation.

**Table 2. Summary of population-level parameter estimates of the Gaussian mixed model.**

| Parameter | Inversion Effect | | | | Negation Effect | | | |
|---|---|---|---|---|---|---|---|---|
| | Median | Mean | CrI lower | CrI upper | Median | Mean | CrI lower | CrI upper |
| Peak Location | 91.53 | 91.52 | 83.40 | 98.56 | 85.92 | 85.99 | 78.01 | 93.31 |
| Standard Deviation | 33.99 | 34.06 | 24.66 | 43.92 | 38.54 | 39.39 | 24.34 | 53.77 |
| Base Amplitude | 1.20 | 1.20 | .99 | 1.42 | .54 | .53 | .25 | .80 |
| Peak Amplitude | 1.10 | 1.11 | .78 | 1.41 | 1.18 | 1.18 | .83 | 1.49 |

The 89% CrI of the Base Amplitude parameter on inversion and negation effects do not overlap indicating a significant difference between the two conditions.

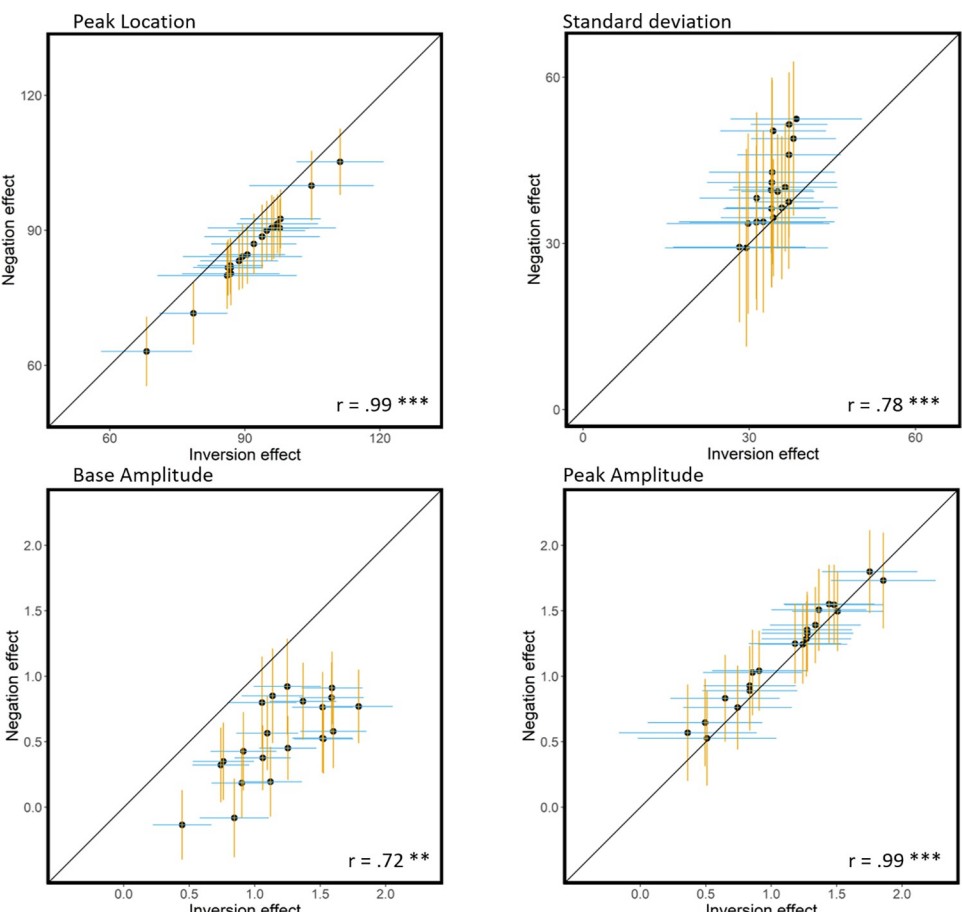

**Fig 5. Scatter plot of the four group(i.e., participant)-level Gaussian-model parameter estimates of inversion and negation effects with standard deviation for each group-level parameter estimates as error bars; in blue for inversion and in yellow for negation.** The four Gaussian parameters group-level estimates are highly correlated between inversion and negation effects, the Pearson correlation coefficients are indicated for each parameter. We find a strong correlation between the Gaussian parameters of the inversion and negation orientation tuning profiles, yet several differences can be observed graphically. While peak location and amplitude are very similar across inversion and negation effects (close to diagonal), the base amplitude individual estimate was largest for the inversion effect in all participants, indicating that the decrease in overall recognition performance was consistently more drastic for inversion than negation. Most group-level estimates of Standard deviation were located above the diagonal, in favor of the negation effect section, indicating a narrower horizontal tuning for inversion than negation. Pearson correlation coefficients are written for each of the four parameters with *** for p < .0001, and ** for p < .001. Using partial correlations controlling for the Natural condition we found no evidence that these high correlations were driven by the use of the Natural condition to calculate the effects (see S3 in S1 File).

Comparison of population-level model parameter estimates between inversion and negation effects highlights their similar tuning to face orientation content (Fig 5). To examine whether the similarity in the Gaussian parameters reflects a functional relationship between inversion and negation effects at the individual subject level, we ran correlation analyses on the group (i.e., participant)-level Gaussian parameters' posterior predictions. Correlations between inversion and negation effects were very high for all four Gaussian model parameters' (all r $\geq$ .72, all p < .001). Stronger correlations were found for Peak location (r = .99, p < .0001, CI = [.99; 1]) and Peak amplitude (r = .99, p < .0001, CI = [.97; .99]) parameter estimates than for Base amplitude (r = .72, p < .001, CI = [.42; .88]) and Standard deviation (r = .78, p < .0001, CI = [.53; .91]; Fig 5) parameter estimates. Supplementary analyses (S3 and S7 in S1

File) indicate that the strong functional dependence of the Gaussian tuning parameters of inversion and negation effects described here is due neither to the use of the upright-positive condition as a common comparison condition (i.e., the correlations were similar when computed while controlling for the variance due to the upright-positive condition; see S3 in S1 File), nor to the use of a smooth Gaussian model to fit the data (i.e., results replicate with a Linear Mixed Model; see S7 in S1 File).

## Discussion

Past evidence showed that the horizontally-oriented cues of the face stimulus enable better identity recognition accuracy than facial cues in other orientation ranges. Yet, the nature of the information contained in the horizontal range remained elusive. Face identity recognition is thought to rely on two main and separable sources of information: shape and surface [50, 51, 65]. The drastic recognition impairments caused by inversion and negation have been attributed to the disrupted processing of shape and surface cues, respectively [44, 53, 55, 66, 67]. Here, we exploited the relative functional selectivity of these effects to characterize the shape versus surface nature of the cues carried by the horizontal range of the face image. Participants identified face images filtered to preserve a narrow orientation band selectively and presented in natural (i.e., upright and in natural contrast polarity), inverted or negated conditions. We estimated the orientation tuning profile of the impairments caused by inversion and negation on face identification performance. As expected, inversion and negation both drastically impaired face identification performance compared to when faces were shown in natural viewing conditions (i.e., upright and in normal positive contrast). Overall, the impairment caused by inversion was stronger than negation. However, the orientation profile of the two effects was strikingly similar: both effects peaked in the horizontal range with similar strength and sharpness. Both effects decreased as the orientation neared vertical. The similar peak location and peak amplitude (which is the difference of effect between its highest, i.e., in the horizontal range, and its lowest, i.e., in the vertical range) indicate that inversion and negation both mostly disrupted access to the horizontally-oriented content of the face stimulus. As inversion and negation prevent the face-specialized processing of shape and surface cues, respectively, these results indicate that the horizontal preference of human face recognition is due to this range yielding a privileged access both to face shape and surface, i.e., two main sources of face identity information.

In line with past evidence, identity recognition in natural viewing condition was tuned to the horizontal range (S4 in S1 File; [16, 17, 19, 20], although compared to the reported results, the horizontal tuning we measured here was broader. Goffaux & Greenwood (2016) [20] found that the bandwidth of the horizontal tuning was influenced by the availability of external facial features such as facial outline cues to perform face recognition. Namely, the horizontal tuning was significantly broader when face outline was available. Our stimulus set contained much larger and more natural outline variations than in this previous study, which could explain the relative broadening of the horizontal tuning. Using familiar faces, as in the present study, Dakin and Watt (2009) [16] also seemed to have found a relatively broadly tuned recognition performance (as they do not report it, this assumption is based on their figure). Finally, to ensure participants performed above chance level in negated and inverted trials, we set the orientation filters to have a broader standard deviation than used in other studies (25˚ compared to e.g. 15˚ in Goffaux & Greenwood (2016) [20]). This orientation filter breadth resulted in some overlap in the oriented energy across adjacent orientation ranges. This may to some extent explain the shallower horizontal tuning in the Natural condition and why recognition performance was close to ceiling in the horizontal and adjacent oblique orientations, (See S4 in S1 File).

To investigate their orientation tuning profile, we submitted the effects of inversion and contrast negation on face identification to a Bayesian non-linear Gaussian mixed model. This analysis showed that both effects peaked in the horizontal range and followed a strikingly similar orientation tuning profile. Our finding of the disruption of horizontal tuning of face identification with inversion effect is in line with previous studies comparing the orientation tuning of upright and inverted faces [19–21]. The disruption of the horizontal tuning with negation we measured however constitutes new knowledge regarding the orientation tuning of face identification.

The only difference between the inversion and negation effects was at the level of the base amplitude of the orientation tuning profile. Combined with the fact that the peak amplitude was as large for inversion as it was for negation, we conclude that inversion impaired recognition performance more strongly than negation overall (i.e., across orientations, see Fig 4a). Studies about contrast negation showed that the presence of pigmentation cues (e.g., skin texture) in the face stimulus increases the strength of the negation effect [52, 53]. Orientation filtering may have reduced the availability of pigmentation cues, and therefore reduced the impact of negation overall. The weaker effect of negation compared to inversion may also indicate that inversion disrupts face perception at an earlier processing stage than negation [45, 47]. Indeed, compared to negation which flips the polarity while preserving the shape and retinotopic location of the contrast variations making up the face stimulus, inversion drastically changes the latter, and may impact visual processing as soon as V1 (see [18] for supportive evidence); such earlier impact may result in an overall stronger impact on behavioral performance of inversion compared to negation.

Through the demonstration of the additivity and functional selectivity of the behavioral and neural effects, past research showed that inversion and negation affect face processing differently [44–48]. This view is suggested by the fact that we have found no evidence for an association between the two effects with full-spectrum faces in our data (see S5 in S1 File). Altogether, our findings indicate that despite inversion and negation showing some functional independence in full-spectrum viewing conditions, they affect access to face-oriented content in a similar way. Previous work has demonstrated that the specialized holistic processing of the face stimulus is affected by inversion and, to a lesser extent, by negation [5, 34, 45, 49, 50, 57], the horizontal advantage over other orientations may therefore arise from the essential nature of the horizontal shape and surface cues to allow holistic face processing.

Another plausible account to the similar orientation tuning properties observed across inversion and negation effects is that the horizontal range of face information carries a more primitive source of information yielding common entry point to the functionally separate shape and surface cue processing. The horizontal structure of a human face has indeed been proposed to convey the natural statistical properties of the face stimulus. These natural statistics are described as an alternance of light and dark horizontal bands along the vertical axis of the face image, from forehead to chin (see Fig 1 of Liu-Shuang et al., 2022 [41], the appearance of which is drastically distorted by inversion and negation [16, 19, 20]. As a matter of fact, the horizontal range of face information was shown to drive the human behavioral and neural responses to faces at more generic levels of categorization than the fine-grained level of identity recognition tested here. A preference for the horizontal was indeed found in infants and children in face categorization [24, 68]. Emotion categorization has also been shown to rely on the horizontal range of face information [26, 27, 69, 70]. At the neural level, Jacques et al., (2014) [22] showed that the face horizontal tuning appears as early as the activation of face specialized-processing (in the N170 time window), in other words as early as face detection. Furthermore, Goffaux et al., (2016) [18] showed a preferential activation of face-preferring brain areas such as the Fusiform Face Area and the Occipital Face area to upright faces containing

horizonal information. In both cases the neural preference for the horizontal was disrupted with inversion.

A recent unpublished study of our lab suggests that the horizontal range not only conveys the statistical regularities of the human face category, but that it also carries stable identity cues that are optimal for viewpoint-tolerant recognition of identity (see also [19]). Supporting the crucial contribution of horizontal information to fine-grained identity recognition, horizontal tuning is positively correlated to the face identification performance [21, 28, 29] and the horizontal tuning is stronger for the recognition of familiar than unfamiliar faces [30]. Past and present evidence therefore indicates that the horizontal advantage may actively operate at various processing stages leading from early detection to fine-grained identification. Future studies need to investigate the orientation profile of human face processing at different processing stages to uncover the potential shifts in the orientation dependence of the neural representation of faces over the course of their processing by the human brain (see [71] for such an approach).

To conclude, we have shown that the disruption of human face identity recognition by inversion and negation follows the orientation preference of face identity recognition: it is horizontally tuned. Although inversion and negation presumably disrupt distinct sources of face information, their orientation tuning profiles are highly similar and correlated. These results suggest that the horizontal tuning of face identification is due to this range conveying the two main sources of information driving the specialized processing of face identity: shape and surface.

## Supporting information

**S1 File. File containing further analysis and explanation.** All figures and analysis cited in the manuscript are in this file.
(DOCX)

## Acknowledgments

We thank Eunice Tshibambe for her help with data acquisition, and M. Umut Canoluk for his help with statistical analysis. This research has benefitted from the statistical consult with Statistical Methodology and Computing Service, technological platform at UCLouvain–SMCS/LIDAM, UCLouvain, via the help of Vincent Bremhorst.

## Author Contributions

**Conceptualization:** Helene Dumont, Alexia Roux-Sibilon, Valérie Goffaux.

**Data curation:** Helene Dumont.

**Formal analysis:** Helene Dumont, Alexia Roux-Sibilon.

**Funding acquisition:** Valérie Goffaux.

**Investigation:** Helene Dumont.

**Methodology:** Helene Dumont, Alexia Roux-Sibilon, Valérie Goffaux.

**Project administration:** Valérie Goffaux.

**Resources:** Valérie Goffaux.

**Supervision:** Valérie Goffaux.

**Validation:** Valérie Goffaux.

**Visualization:** Helene Dumont.

**Writing – original draft:** Helene Dumont, Valérie Goffaux.

**Writing – review & editing:** Helene Dumont, Alexia Roux-Sibilon, Valérie Goffaux.

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
