## [Decision Letter · Decision Letter 0]

7 Jun 2024

PONE-D-24-18307

Horizontal face information is the main gateway to the configuration and surface cues to familiar face identity.

PLOS ONE

Dear Dr. Dumont,

Thank you for submitting your manuscript to PLOS ONE. After careful consideration, we feel that it has merit but does not fully meet PLOS ONE’s publication criteria as it currently stands. Therefore, we invite you to submit a revised version of the manuscript that addresses the points raised during the review process.

Your manuscript has been reviewed by two experts in the topic. As you will see below, both reviewers were positive about your work, but they also made several clear and constructive recommendations for further improving the manuscript which I would encourage you to implement in the revision. One point that came up in both reviews is that the relationships between face inversion/configuration cues and between negation/surface cues might not be as straightforward as it is claimed in the manuscript, and I agree with the reviewers that a more neutral framing of the study (possibly including in the title) might therefore be necessary.

We look forward to receiving your revised manuscript.

Kind regards,

Patrick Bruns

Academic Editor

PLOS ONE

Journal Requirements:

Reviewers' comments:

Reviewer's Responses to Questions

**Comments to the Author**

1. Is the manuscript technically sound, and do the data support the conclusions?

Reviewer #1: Yes

Reviewer #2: Yes

2. Has the statistical analysis been performed appropriately and rigorously? 

Reviewer #1: Yes

Reviewer #2: Yes

3. Have the authors made all data underlying the findings in their manuscript fully available?

Reviewer #1: No

Reviewer #2: Yes

4. Is the manuscript presented in an intelligible fashion and written in standard English?

Reviewer #1: Yes

Reviewer #2: Yes

5. Review Comments to the Author

Reviewer #1: This paper examines whether horizontal tuning for faces provides similar access to configural and surface cues. Participants performed a same-different task with familiar faces across various orientation filters on natural, inverted, or negated images. The authors conclude that horizontal tuning is equally important for configural cues and surface cues. This work extends a growing literature on horizontal tuning for faces. It replicates prior inversion data and contribute new negation data. I don’t have major comments but I have some questions and suggestions that I hope can improve the paper.

1. Upright accuracy is near ceiling at 93%. What are the accuracies for inverted and negated? How far from floor are they? While the choice to analyse d-prime makes sense, an additional analysis of accuracy would help readers better appreciate the data.

2. Mike Burton and others have shown that understanding how different images of the same person can vary (telling faces together) is just as important as understanding image differences between people (telling faces apart). I think the authors can contribute to this issue by analysing same vs different trials separately and testing for potential interactions with orientation filter, image manipulation, or both.

3. Neither configural or surface cues were directly manipulated, and the relationships between inversion/configuration and between negation/surface are not straightforward (see Sormaz et al 2013 for example). I suggest using inversion and negation in the title and discussing the configural and surface cues in the intro and discussion.

4. Supplementary analysis 5 is useful but the current interpretation is based on an indirect contrast between a significant correlation (.61) and a non-significant one (.41). These correlations might not differ statistically so the authors need to compare them directly (see http://quantpsy.org/corrtest/corrtest2.htm) and adjust their interpretations if needed.

Reviewer #2: The authors present novel research on the importance of horizontally oriented facial information for familiar face recognition by showing how this information relates to two seminal effects in the face processing literature: face inversion effect and contrast reversal effect. The first occurs when faces are viewed upside-down, and the second occurs when faces are viewed as a film negative. In the literature, face inversion is often understood to alter processing of configuration cues (e.g., how individual features relate to one another), while contrast reversal is often understood to alter surface cues (e.g., how 3D shape is inferred from a 2D image). The authors show that both image manipulations, i.e., face inversion and contrast reversal, entail a substantial and rather selective drop in sensitivity to horizontally oriented features. Thus, the authors argue, horizontal tuning of the visual system might emerge as the system attunes to configuration and surface cues.

I have a mostly favourable opinion of this work, which has generally sound methodology and statistical analyses. However, I do take issue with some interpretations and general framing of the paper (immediately below). I also have a few requests for clarification and recommendations.

The authors begin with a claim that the cause for human reliance on horizontal facial information is not well understood and that the visual system adapting to configuration and surface cues might explain this (e.g., lines 24-25). This framing hinges on 1) face inversion and contrast reversal effects being valid measures of each construct, and 2) on each construct being empirically distinct. However, disruption of configural processing is only one possible explanation—though one often shared—for the face inversion effect, for instance. But there are alternatives. Furthermore, the authors support their independence hypothesis with the results of a correlation between face inversion and contrast reversal effects for full spectrum faces being nonsignificant. While I appreciate that, strictly speaking, a correlation of r = 0.43, p = 0.051, is not statistically significant, this is not exactly evidence of their independence. Furthermore, the fact that orientation profile modeling parameters of the face inversion and contrast reversal effects correlated almost perfectly suggests these processes do share some fundamental aspects—though admittedly this correspondence was likely inflated vs. the true correlation, due to the smooth nature of Gaussian fitting (see also below). Nevertheless, my criticism essentially boils down to this: Configuration and surface cues are theoretically loaded constructs that are assumed to be altered in the face inversion and contrast reversal effects respectively. Prefacing horizontal tuning with these does nothing to enrich the conversation—and potentially even derails it in light of the points I raised above. An alternative and perhaps better framing is that researchers have long been trying to explain face inversion and contrast reversal effects. Though a disruption of configuration and surface cue processing are possible explanations, there are alternatives. With the recent surge in studies suggesting horizontal tuning is a core feature of the face processing architecture, the authors verified whether and how face inversion and contrast reversal disrupt orientation tuning. This more neutral framing benefits from its lack of assumptions regarding face inversion and contrast reversal effects and adds to a growing literature that shows the functional importance of horizontally orientated information for various aspects of face processing.

Writing is generally good, but the authors should watch out for overreliance on the word “the,” which at times makes reading arduous. E.g., in the sentence starting at line 289, there are 8 “the” in a single sentence, several of which are superfluous. “The purpose of this study was to understand the extent to which the horizontal tuning of face identification reflects the contribution of feature configuration or of the surface and shading of the face image through the comparison of inversion and negation effects on the Orientation filter dependence of identity recognition performance.”

Line 182: Were mean luminance and RMS contrast set before or after applying the elliptical aperture? Please make this more explicit in text.

Line 216: Were actor identities selected on a participant basis? Please make this more explicit in text.

Line 261: It might benefit the reader to remind what chance level was here.

Line 305: The authors should explain why they chose to analyze normalized d’, rather than simply d’. Also, they should provide a bit more detail about how this was done, as I found myself a bit confused by their description of the procedure.

Line 333: Base amplitude and peak amplitude should be clearly explained here (perhaps put in relation with the data at hand). Peak amplitude is also poorly explained at line 440.

Line 469: The authors should consider the possibility that these coefficients are somewhat inflated due to the (smooth) Gaussian fit. They could perhaps exploratorily look at the relationship between inverted and reversed d’ as a function of orientation (perhaps as a random effect). They may also want to correlate individual full spectrum d’s with orientation d’s to look at whether one orientation is more likely to account for individual differences in performance vs. others. I am fully aware the study may be underpowered if the correlations are only small/moderate, but it might nevertheless be informative about some trend within the data (perhaps mixed effects modeling would again offset some of that).

Figure 3: plotting face inversion effect and contrast reversal effect as “normalized sensitivity (d’)” is a bit counterintuitive as it might initially suggest that sensitivity is highest with horizontal information. This could be improved by e.g., also specifying on the axis that a positive d’ value = sensitivity loss, or by plotting orientation profiles upside-down (i.e., with negative d’s).

Figures 3 and 4: should their order be inverted? It seems raw results (Fig 4) should be presented before fitted curves (Fig 3).

Full spectrum trials might provide valuable context, e.g., by being plotted alongside orientation d’, or even plotting orientation d’ as a proportion of the full spectrum d’ (or again, to investigate individual differences across orientations, see above point).

I think upright fitted orientation profiles belong in the main text rather than supplementary materials.

Credible interval: value used (89%) should be in text (Data and analyses).

Regarding the broad orientation tuning found (especially for upright faces), the authors report having used a 25deg wide orientation filter. According to my calculations, adjacent orientation samples (e.g., 90deg and 120deg) were not independent with this width parameter, as they intersect at around mu +/- 15deg. Briefly, a horizontal (90deg) filter and oblique-horizontal (120deg) filter both sample 105deg information at approx. 37% of the maximum sampling proportion. The 120deg filter also samples 90deg information at 2% of the maximum sampling proportion (and inversely, the 90deg filter samples 120deg information at 2%). In fact, their respective areas overlap by about 8%. This is not a problem in and of itself, but it should be recognized by the authors, and it might partially account for somewhat wider orientation tuning found.

Supplementary materials, line 77-78 : “A hypothesis about the origin of the asymmetry is that it would result from faces statistics that often appear tilted and not perfectly upright.” Looking at Figure 1, it seems 60deg and 120deg locally enhance eye, eyebrow and mouth contrast more vs. the 90deg filter, at least subjectively. Could this be a factor, seeing as those are very important features for face processing? Also, is it possible this was caused by how luminance and RMS contrast were equated?

6. PLOS authors have the option to publish the peer review history of their article (what does this mean?). If published, this will include your full peer review and any attached files.

Reviewer #1: No

Reviewer #2: No

---

## [Author Response · Author response to Decision Letter 0]

16 Aug 2024

Dear Dr Bruns,

We have revised our manuscript entitled “Horizontal face information is the main gateway to the shape and surface cues to familiar face identity” (previous title “Horizontal face information is the main gateway to the configuration and surface cues to familiar face identity”) according to the comments and suggestions of the Reviewers who evaluated our work. The revised manuscript and the detailed response to the Reviewers are enclosed. As we revised the manuscript extensively, keeping track of all the changes to the manuscript automatically results in a very messy document. We have therefore decided to highlight the most significant revisions in the Revised Manuscript with Track Changes.docx document by underlining the relevant sections. We briefly summarize our replies and revisions below (detailed replies can be found in the Response to Reviewers.docx).

Following both reviewers’ advice, we toned down the theoretical framing of our study and, in particular, the assumed link between the inversion effect and face configural processing. The revised version stays closer to what can be asserted with greater confidence about inversion and negation: inversion turns edge-related cues upside-down whereas negation flips the luminance gradient of the face surface and preserves face edges. We revised the Title, Abstract, Introduction and Discussion accordingly. 

The Reviewers also pointed us to the fact that our evidence for concluding (in our Supplementary analysis 5) about a 

functional independence of the effects of inversion and negation in the full spectrum picture condition was rather weak (i.e., based on a non-significant correlation but with a p-value of .051). We therefore corroborated our initial correlation analysis by adding (1) a Bayesian correlation analysis including a Bayes factor, and (2) the confidence intervals of the frequentist and Bayesian correlation coefficients, both of which included zero. Overall, we now conclude that there is insufficient evidence from our data to conclude a correlation between the two effects and acknowledge that this may be due to a lack of statistical power.

Following Reviewer 1’s comments, we fully re-analyzed our data based on accuracy instead of the sensitivity index d’. This analysis gave highly comparable results than our original d’ analysis. As we agree with the Reviewer that the accuracy data may still be if interest for readers, we added this analysis in the Supplementary Material. Reviewer 1 also suggested to analyze same and different trials independently, which is a very interesting view of our experimental design. We report this analysis in our response to the Reviewer but not in the manuscript, as it is unfortunately too underpowered to be meaningful

Following Reviewer 2’s suggestions, we added some clarifications to our Methods, data analysis, and supplementary sections, as well as to our Figures. 

Overall, the comments of the two Reviewers allowed us to make significant improvement to the manuscript by clarifying elements of both the Methods and the theoretical foundation of the study. We thank both Reviewers for their helpful and constructive comments, and we hope that the manuscript is now more suitable for publication in your journal.

Yours sincerely,

Hélène Dumont, And on behalf of Alexia Roux-Sibilon, and Valérie Goffaux

---

## [Decision Letter · Decision Letter 1]

17 Sep 2024

Horizontal face information is the main gateway to the shape and surface cues to familiar face identity.

PONE-D-24-18307R1

Dear Dr. Dumont,

We’re pleased to inform you that your manuscript has been judged scientifically suitable for publication and will be formally accepted for publication once it meets all outstanding technical requirements.

Kind regards,

Patrick Bruns

Academic Editor

PLOS ONE

Additional Editor Comments (optional):

The revised manuscript has been reviewed by original Reviewer #1, and I served as second reviewer because original Reviewer #2 was not available at this time. I agree with Reviewer #1 that all points raised during the review process have been thoroughly addressed.

During production of your article, you might want to integrate the supporting figures into the supporting text document (rather than having only the supporting figure captions in the text document), as this will make the document more accessible. Please also use proper item descriptions in the main text, i.e. refer to items as "Supporting Information S1", "Supporting Figure S2" etc. rather than "Supplementary 1" as in l. 168

Reviewers' comments:

Reviewer's Responses to Questions

**Comments to the Author**

1. If the authors have adequately addressed your comments raised in a previous round of review and you feel that this manuscript is now acceptable for publication, you may indicate that here to bypass the “Comments to the Author” section, enter your conflict of interest statement in the “Confidential to Editor” section, and submit your "Accept" recommendation.

Reviewer #1: All comments have been addressed

2. Is the manuscript technically sound, and do the data support the conclusions?

Reviewer #1: Yes

3. Has the statistical analysis been performed appropriately and rigorously? 

Reviewer #1: Yes

4. Have the authors made all data underlying the findings in their manuscript fully available?

Reviewer #1: Yes

5. Is the manuscript presented in an intelligible fashion and written in standard English?

Reviewer #1: Yes

6. Review Comments to the Author

Reviewer #1: (No Response)

7. PLOS authors have the option to publish the peer review history of their article (what does this mean?). If published, this will include your full peer review and any attached files.

Reviewer #1: No

---

## [Editor Report · Acceptance letter]

27 Sep 2024

PONE-D-24-18307R1 

PLOS ONE

Dear Dr. Dumont, 

I'm pleased to inform you that your manuscript has been deemed suitable for publication in PLOS ONE. Congratulations! Your manuscript is now being handed over to our production team.

Kind regards, 

on behalf of

Dr. Patrick Bruns 

Academic Editor

PLOS ONE